# Cognitive Decline Related to Diet Pattern and Nutritional Adequacy in Alzheimer’s Disease Using Surface-Based Morphometry

**DOI:** 10.3390/nu14245300

**Published:** 2022-12-13

**Authors:** Hua-Tsen Hsiao, Mi-Chia Ma, Hsin-I Chang, Ching-Heng Lin, Shih-Wei Hsu, Shu-Hua Huang, Chen-Chang Lee, Chi-Wei Huang, Chiung-Chih Chang

**Affiliations:** 1Department of Nursing, National Tainan Junior College of Nursing, Tainan 700007, Taiwan; 2Department of Statistics and Institute of Data Science, National Cheng Kung University, Tainan 701401, Taiwan; 3Department of Neurology, Cognition and Aging Center, Institute for Translational Research in Biomedicine, Kaohsiung Chang Gung Memorial Hospital, Chang Gung University College of Medicine, Kaohsiung 833401, Taiwan; 4Center for Artificial Intelligence in Medicine, Chang Gung Memorial Hospital, Taoyuan 333008, Taiwan; 5Bachelor Program in Artificial Intelligence, Chang Gung University, Taoyuan 333323, Taiwan; 6Department of Radiology, Kaohsiung Chang Gung Memorial Hospital, Chang Gung University College of Medicine, Kaohsiung 833401, Taiwan; 7Department of Nuclear Medicine, Kaohsiung Chang Gung Memorial Hospital, Chang Gung University College of Medicine, Kaohsiung 833401, Taiwan

**Keywords:** Alzheimer’s disease, cognitive decline, diet pattern, blood biochemistry, cortical thickness

## Abstract

Dietary pattern (DP) results in nutrition adequacy and may influence cognitive decline and cortical atrophy in Alzheimer’s disease (AD). The study explored DP in 248 patients with AD. Two neurobehavioral assessments (intervals 13.4 months) and two cortical thickness measurements derived from magnetic resonance images (intervals 26.5 months) were collected as outcome measures. Reduced rank regression was used to assess the groups of DPs and a linear mixed-effect model to explore the cortical neurodegenerative patterns. At screening, underweight body mass index (BMI) was related to significant higher lipid profile, impaired cognitive function, smaller cortical thickness, lower protein DP factor loading scores and the non-spouse caregiver status. Higher mini-mental state examination (MMSE) scores were related to the DP of coffee/tea, compared to the lipid/sugar or protein DP group. The underweighted-BMI group had faster cortical thickness atrophy in the pregenual and lateral temporal cortex, while the correlations between cortical thickness degeneration and high HbA1C or low B12 and folate levels were localized in the medial and lateral prefrontal cortex. The predictive model suggested that factors related to MMSE score were related to the caregiver status. In conclusion, normal or overweight BMI, coffee/tea DP group and living with a spouse were considered as protective factors for better cognitive outcomes in patients with AD. The influence of glucose, B12 and folate on the cortical degeneration was spatially distinct from the pattern of AD degeneration.

## 1. Introduction

Neurodegeneration is a key feature in Alzheimer’s disease (AD) that leads to cognitive decline and dementia [1]. Many observational studies have linked dietary pattern (DP), metabolic factors and gray matter atrophy for cognitive decline predictions [2,3,4,5,6,7,8,9]. In Asia, traditional Chinese food has been associated with overweight, hypertension and greater cognitive decline in the elderly [10]. However, DP has evolved with advances in education and globalization. It is important to update the food-preference categories and assess their association with cognition for public brain health recommendations [11].

The associations between DP and cognitive functions have been extensively explored in the elderly. In short, the favorable DPs include higher intake of vegetables, fruits, legumes, fish, lean red meat, nuts, poultry, olive and seed oils, which are rich in antioxidants, flavonols, monounsaturated fatty acids and polyunsaturated fatty acids [11,12,13,14,15,16,17,18]. Daily intake of green leafy vegetables rich in β-carotene, folate, lutein, and phylloquinone may slow the age-related cognitive decline [19]. Higher adherence to a Mediterranean diet has been associated with a lower incidence and slower cognitive decline of dementia [12,20,21,22,23]. The Mediterranean-DASH diet Intervention for Neurodegenerative Delay diets have demonstrated benefits in AD cognitive reserve [23]. In contrast, intake of red meats, high-fat dairy, cream, butter, processed meat and sugar-sweetened beverages were considered as harmful DPs for dementia [10,21,24,25,26,27]. Another report suggested lower vegetable intake, higher alcohol and higher trans-fat intake have been associated with poorer memory or executive functioning in AD [28]. DP is highly culture-driven and can be considered as a modifiable factor. The relationships between DPs and cognitive decline in patients with AD require more data.

One of the most promising tools that helps to monitor the structural changes in time is magnetic resonance imaging (MRI). Various MRI processing toolboxes have been developed to measure the structural changes in degenerative disorders. The parcel morphometry matrix, such as cortical thickness, has been proven valid in stratifying AD [29] or subtypes of mild-cognitive-impairment groups [30]. In Surface-based analysis, the cerebral cortex is modeled as a 2D sheet, from which thickness, surface area and volume can be computed at each point. These quantities can be analyzed in a group surface space after surface-based registration and surface-based smoothing. Surface-based methods exhibit superior performance to volumetric approaches for voxelwise analysis and provide precise localization of functional foci and inter-subject registration. Previous work has validated the consistency between cortical thickness estimations by FreeSurfer (http://surfer.nmr.mgh.harvard.edu accessed on 15 November 2022) [31,32] and histological measurement [33]. The linear mixed-effects (LME) model includes random effects, which account for the intra-visit correlations in longitudinal data [34,35,36,37,38]. The LME model implemented in FreeSurfer has enabled the handling of unequal timing and different number of time points across the subjects. It also models the temporal correlation and different variances across measurement. The use of the spatiotemporal LME method based on a parcellation of the brain into homogenous area constructed the full spatiotemporal covariance structure by assuming a common temporal covariance structure across all the points with a simple spatial covariance structure [36]. This solves the random effects and the covariance structure of the errors.

Nutritional status and blood profiles, such as blood sugar, vitamin B12, folic acid and lipid profiles, represent the downstream of DP that can be monitored [39,40]. The survey of DP, in terms of food preference, validated whether these DPs were associated with blood profiles and the adequacy of nutritional status in AD may help to understand the influential factors for interventions. Based on the findings of cortical thickness and risk factors in AD, the cortical thickness can be considered as an endophenotype of AD pathophysiology. An indirect link between hyperglycemia and cognitive decline was found to be mediated by the lower cortical thickness [41]. Higher homocysteine levels were associated with lower cortical thickness in several brain regions [42] while the intake of total vitamin D was associated with higher cortical thickness in the temporal lobes [43]. Controversies exist in the Finnish Geriatric intervention study, in that cortical thickness or white matter lesions were not significantly different in the intervention and control groups [44].

Factors related to DPs may show complex interactions. The specific aim of this study was to explore the significant demographic and metabolic factors that reflect DP in patients of AD and assessed whether the DP may influence cognitive scores via changes in these parameters.

## 2. Materials and Methods

### 2.1. Study Design

The clinical data were retrieved from the Cognitive and Aging (CAC) database. The CAC database was from a longitudinal clinical and MRI cohort study focusing on elder subjects with memory complaints. Patients enrolled in the CAC database received annual check-ups at the department of general neurology, Chang Gung Memorial Hospital, Kaohsiung [45]. All patients had annual cognitive assessments and the structural MRI was performed on a 2-year basis. Participants in this DP exploratory study gave a separate consent for the Dietary assessment form. Ethics committee approval was obtained at Chang Gung Memorial Hospital, Kaohsiung. Written Informed consent was obtained from all participants and legally authorized representatives in the cases with cognitive impairment.

### 2.2. Inclusion and Exclusion Criteria

The DP exploratory study enrolled patients with early-stage AD who were diagnosed according to the International Working Group-2 criteria [46] and further confirmed by amyloid imaging if the consensus panel did not agree on the diagnosis. Only those with clinical dementia rating (CDR) score [47] ≤1 were enrolled. In addition, subjects of AD were only eligible if they had been followed-up at least 3 years prior to the DP exploratory study and in a stable condition under acetylcholine esterase inhibitor treatment from the time of diagnosis. We defined the rapid cognitive decline (RCD) [48] by mini-mental state examination (MMSE) decline ≥3 year within the follow-up period. The exclusion criteria were a past history of clinical stroke, a negative amyloid scan, a modified Hachinski ischemic score >4 and depression. After checking the inclusion and exclusion criteria, a total of 248 patients with AD (109 males, 139 females) was included in this DP exploratory study.

### 2.3. Demographic Data

The demographic data included gender, years of education, *ApoE4* status, body mass index (BMI) [49], marital status, self-care ability, comorbidity and major caregiver status (spouse or others). For education, we considered it as a non-modifiable factor as our patients with AD may not obtain further educational years at the time of survey. *ApoE4* carriers were defined as participants with one or two E4 alleles. We classified BMI into three groups: underweight (BMI < 18.5 kg/m^2^), normal (BMI 18.5–22.9 kg/m^2^), overweight (BMI 23–24.9 kg/m^2^) and obese (BMI ≥ 25 kg/m^2^) [50]. The definitions of hypertension [51], type II diabetes [52] and hyperlipidemia [53] were according to the reported criteria.

### 2.4. Dietary Assessment and Dementia Functional Survey

The food frequency questionnaire (FFQ) was conducted by trained interviewers at enrolment to assess dietary intake in the AD patients. As the patients may have memory gaps, the information of FFQ was obtained from the major caregiver. The 22 food groups and food items included in this study are shown in Appendix A. The frequency of intake was measured using the following seven categories: never or less than 1 time per month, 1–3 times per month, 1–2 times per week, 3–4 times per week, 5–6 times per week, 1–2 times a day and more than 3 times a day.

For functional survey, we used the MMSE, CDR [47], Mandarin version of the Every Day Cognition (ECog) scale [54], Neuropsychiatric Inventory (NPI) [10,55] and Frontal Behavior Inventory (FBI). The ECog scale is a validated informant-rated questionnaire based on one global factor and six domain-related factors [56]. The psychometric properties in the ECog scale focus on mild symptoms in everyday function and cognition, and it is suitable to evaluate early-stage AD. The Mandarin version of ECog has been validated [54] to reflect brain metabolism status in early-stage AD. The NPI evaluates 12 neuropsychiatric disturbances common in AD and evaluates caregiver stress in relation to these symptoms. The FBI is a 24-item, quantifiable questionnaire directed at the caregiver [57].

We checked the blood chemistry profile of each subject at enrolment, including hemoglobin A1c (HbA1c), high-density lipoprotein (HDL), low-density lipoprotein (LDL), total cholesterol, triglyceride, vitamin B12 and folate levels.

### 2.5. Cognitive Outcomes (Baseline and One Year Prior to Enrolment)

Two cognitive measurements were used for outcome analysis. The MMSE^a^ corresponded to the time point at DP screening while the MMSE^b^ was the score one year prior to the screening. The mean interval between MMSE^a^ and MMSE^b^ was 13.4 months.

### 2.6. MRI Acquisition, Salient Regions of Interest and Composite Cortical Thickness

Each subject received a baseline T2-weighted MRI scan to confirm the absence of organic lesions in the brain. White matter hyperintensities of Fazekas scale >2 were excluded. Thus, 3DT1 MR images were obtained using a 3T GE Discovery 750 (GE Medical Systems, Milwaukee, WI, USA) and were acquired using a T1-weighted, inversion-recovery-prepared, three-dimensional, gradient-recalled acquisition in a steady-state sequence (repetition time (TR) = 12.24 ms; echo time (TE) = 5.18 ms; field of view = 256 × 256; matrix size = 256 × 256; number of excitations (NEX) = 1; inversion time (TI) = 450 ms; flip angle = 15) with a 1 mm slice sagittal thickness with a resolution of 0.5 × 0.5 × 1 mm^3^. All MRI scans were preprocessed on the same workstation (Macintosh iMac Pro 2017, macOS Catalina, version 10.15.16) using FreeSurfer image analysis suite v7.1.1 http://surfer.nmr.mgh.harvard.edu (accessed on 15 November 2022) for cortical parcellation (Appendix B). For each 3D-T1 image, we performed the longitudinal streams, which segmented all serial timepoints at once to increase the longitudinal stability.

The hippocampus, amygdala, and nucleus accumbens of each hemisphere represented salient regions of interest and the cortical thickness was calculated using the linear combinations of these regions, adjusted for estimated total intracranial volume (eTIV). We included 2 time points of MRI in the longitudinal cortical thickness model. The cortical thickness^1^ represents the time at screening while the cortical thickness^2^ represents the closest time point with cortical thickness^1^. The mean interval between cortical thickness^1^ and cortical thickness^2^ was 26.5 months.

Meanwhile, we also retrieved all available MRI data from the CAC cohort for the time-effect modeling. The MRI intervals ranged from 1.5 years to 2 years and the follow-up duration from 4 years to 10 years. The spatiotemporal linear mixed-effect (LME) model was performed in MATLAB 2019b (The Mathworks Inc., Natick, MA, USA) [36]. The LME model considered cortical thickness as a dependent variable and age, gender, education year and eTIV as the nuisance covariates. The interaction between diagnostic groups (all patients, RCD, stable group) and disease duration (months) was modeled. The interactions in BMI groups (overweight-normal or underweight-normal) and disease duration (months) were performed to assess the slope changes compared with normal BMI group [49]. For the effects of blood chemistry profiles, we performed correlation analysis with HbA1c, total cholesterol, HDL, LDL, B12 and folate as independent variables. With cluster-wise correction computed with parametric Gaussian-based simulations to compute the false-positive rate 0.05, we used a vertex-wise threshold of 3.0 [58].

### 2.7. Statistical Analysis

We used the Statistical Package for the Social Sciences (SPSS Statistics for Windows, Version 17.0, SPSS Inc., Chicago, IL, USA) for all statistical analyses. For descriptive analysis, categorical variables were reported as percentages and continuous variables were reported as mean ± standard deviation (SD). We assessed the differences in proportions and means using the chi-square (χ^2^) test, Fisher’s exact test, Mann–Whitney U test and Kruskal–Wallis test. SAS statistical software (version 9.4, SAS Institute, Cary, NC, USA) was used for reduced rank regression (RRR) analysis. RRR was used to group and reduce the dimensionality of the 22 food intake frequencies and to identify the direction of the largest correlation coefficient with the MMSE^a^. The DP scores in each group were calculated. We used multiple regression model and the MMSE^a^ score was considered as the response variable. The independent variables included significant factors from the demographic factors and the 3 DP scores. For those showing high collinearly, we used the least absolute shrinkage and selection operator (LASSO) regression model to identify the coefficients of factors and their 95% confidence intervals (CIs) related to MMSE^a^ scores and set the statistical significance level at *p* < 0.05 (two-tailed).

## 3. Result

### 3.1. Factor Loading of 22 Food Frequencies

According to the RRR (Table 1), there were three major DP groups while the DP scores indicated the weighted sum of standardized food-intake frequencies. The factor loadings showing absolute values ≥ 0.5 were: (1) protein group: lean meat, skimmed milk; (2) coffee/tea group: coffee/tea; and (3) lipid/sugar group: entrails, sugar. Among these, the coffee/tea group had the highest factor loading scores to MMSE.

### 3.2. Gender Differences in BMI and DPs

Demographic data showing differences in gender are listed in Table 2. Most of the men were married, in the overweight-BMI group and lived with their spouse. The male gender had significantly higher MMSE^a^ and MMSE^b^ scores, educational level and factor scores in the coffee/tea group. The cortical thickness did not show a gender effect.

### 3.3. Underweight-BMI Had Lower Cognitive Performance, Smaller Composite Cortical Thickness and Higher Lipid Profiles

We stratified the patients into four BMI groups. The underweight-BMI group showed the lowest cortical thickness compared with the other two while they also had the lowest MMSE scores (Table 3). Meanwhile, the underweight-BMI group had higher HDL, LDL and cholesterol levels, but lower HbA1c and triglyceride levels. The DP in the overweight and obese groups showed higher protein group factor scores and their major caregivers were the spouse.

### 3.4. Factors Related to RCD

While the MMSE^b^ scores were not significantly different between RCD and stable group, scores of MMSE^a^ were significantly lower in the RCD group (Table 4). Factors associated with RCD were the underweight to normal BMI status, higher FBI scores, dependent self-care ability and presence of hyperlipidemia.

### 3.5. Modifiable and Non-Modifiable Factors Associated with Cognitive Performance

We constructed a regression model (Figure 1) using significant factors. Because MMSE^a^ and the functional scores were pair-wise dependent, we used MMSE^a^ as the major cognitive outcome. Because gender was related to caregiver status, self-care ability, education level and BMI, we used the LASSO regression model to stabilize the coefficients.

Table 5 shows that age, age*age, gender, exercise, education and a factor score of 3 DPs were significantly associated with MMSE^a^ score. We further divided the data according to the major caregiver status. For those not living with spouse, educational year related to higher MMSE^a^ scores. In contrast, male gender, exercise, educational year and the factor scores of the protein group and lipid/sugar group were significantly associated with a higher MMSE score for the patients living with a spouse. Meanwhile, the scatter plot between age and MMSE^a^ showed different trendlines for those “living with a spouse” or those “living with others” (Figure 2).

### 3.6. Spatiotemporal Cortical Degenerative Patterns

We assessed the cortical degenerative patterns stratified by clinical subtypes, BMI groups and blood levels (Figure 3). Overall, the AD patients showed degeneration in the medial and lateral temporal regions and pregenual areas (Figure 3A). The RCD group showed more extensive cortical thickness atrophy in the lateral temporal region and medial temporal regions, but statistical significance was not found compared with the stable group. Although a higher proportion of the patients was obese, the effects of cortical degeneration were more obvious in the underweight group (Figure 3B) and showed in the pregenual region and lateral temporal area. For blood profile correlations (Figure 3C), a higher HbA1c was related to medial prefrontal region degeneration. The effects of B12 and folate were emphasized in the anterior cingulate and lateral temporal regions. Regions related to HDL and total cholesterol were relatively scattered, while the correlations with LDL or total triglyceride levels were not significant.

## 4. Discussion

### 4.1. Major Findings

In AD, our regression model explored DPs and other clinical factors (socio-demographic and metabolic factors) that predicted general cognition or surface morphometry at baseline or at a one-year interval. There were three major findings. First, we reported significant modifiable and non-modifiable factors related to cognitive test scores in AD (Figure 1). The significant modifiable factors included DPs, BMI and exercise habits, while the non-modifiable factors were caregiver status, educational year, age, gender and *ApoE4* carrier status. Among these, the caregiver status was considered as non-modifiable factor that may influence the outcomes via the interactions with exercise and beneficial DPs. Second, the cognitive test scores showed a high correlation with the composite cortical thickness measurements using hippocampus, amygdala and nucleus accumbens. The finding supported the important endophenotypic regions for estimating cognitive performances in AD. Lastly, we found that the underweight-BMI group had a faster cortical thickness atrophy in the lateral temporal, hippocampus and anterior cingulate gyrus. As these areas colocalized with the atrophic regions in the RCD group, the nutritional status may exert its clinical significance on the degenerative network of AD. In contrast, regions correlated with HbA1C, B12 and folate (Figure 3C) were mainly found on the medial frontal and dorsolateral frontal regions, supporting other neurobiological mechanisms of these metabolic factors in AD pathology.

### 4.2. DP-Related Factors and Cortical Atrophy

In this study, we found two important categories (BMI status, blood profiles) that were related to cortical degeneration in AD. It is worth pointing out that the LME model here was based on the whole brain morphometry and all the available historical MRI data in our database. Therefore, the regions reported here represented longitudinal degenerative pattern rather than a cross-sectional observation of the two categories.

BMI is a clinical index for nutritional adequacy [59]. Shaw et al. assessed the relationships between BMI and cortical thickness areas in 405 midlife and 398 late-life normal cognitive individuals [60]. In the midlife individuals, increasing BMI was associated with greater cortical thinning in the posterior cingulate cortex. In late life, increased BMI was associated with right supramarginal cortex and frontal cortical thickness. Meanwhile, decreased BMI was also associated with right caudal middle frontal cortex cortical thickness. Their study indicated paradoxical relationships between BMI and cortical thinning during aging. High BMI at midlife was associated with dementia susceptibility while in late life, high or low BMI could both lead to regional cortical thickness changes. In other studies enrolling cognitively normal late-life individuals, high BMI status was associated with lower cortical amyloid β [61], and a male-related BMI effect was found [62]. To our understanding, most of the studies exploring BMI and cortical thickness were performed in normal cognitive individuals. The link among nutritional adequacy, DPs and surface morphometry in patients with AD was not reported before.

Our LME model provided the relationships between BMI and brain degenerative patterns (Figure 3B) in patients with AD. The underweight-BMI group had greater volume atrophy in the temporal and the sub-genual regions, compared with the normal BMI group, and these regions also colocalized with the degenerative process in AD (Figure 3A). Based on our study results, underweight BMI was related to faster cortical atrophy in AD. Previously, Hsiao et al. [49] reported the nutrition status and adequacy of BMI ranges in patients with AD. Considering the balance of comorbidities and nutritional status, a BMI of 22–23 kg/m^2^ was suggested in AD. Our underweight-BMI group was defined as 18.5 kg/m^2^ and our study results advise increments in BMI to avoid underweight status for a better cognitive outcome.

Meanwhile, our study also explored the linear relationships between cortical thickness and blood profiles in AD. We found greater cortical thickness progression occurred in higher HbA1C, lower B12 and folate (Figure 3C). As these regions were in the medial frontal regions and were spatially distinct from the AD degenerative areas (Figure 3A), we speculated the mechanisms of high HbA1C status or low B12 and folate to neurodegenerations were distinct from the amyloid or tau trajectories. As these three blood profiles were related to disease progression and were considered as modifiable, proper monitoring of these three blood profiles should be emphasized in clinical practice.

BMI and blood profiles were tightly linked and our four BMI groups were associated with different blood profiles and DPs. According to our results, the underweight group had higher cholesterol but lower triglyceride and protein factor loadings. Although the impact of lipid profiles to cortical thickness was minor compared with HbA1C, the balance and proper supply of lipid/sugar DPs in the underweight group should be closely monitored.

### 4.3. DP and Cognitive Functions in the AD Patients

Our DP factor analysis suggested a beneficial or clustering relationship between food preferences and MMSE. The mixed patterns of food preferences suggest flexible modifications on educational plan. DPs are highly culture related, especially in Asia. One study in Taiwan assessed DP and domain-specific cognitive function in 475 elderly subjects aged ≥ 65 years and found that logical memory-recall results were associated with a higher vegetable DP, while a meat DP prevented a decline in attention [26]. Two studies in Korea explored 765 and 806 elderly subjects, respectively [63,64], and found that the white-rice-only DP group [64] and white rice, noodles, and coffee DP group [63] declined faster. In one Japanese community study (*n* = 1006, 60~79 years), a DP related to plant foods and a lower amount of rice was related to a lower risk of dementia [17]. Another Japanese study reported results like ours, in that they found that a DP with soy products improved cognitive function in 635 elders aged 69~71 years [65]. The DPs in the studies may have different names; however, they may reflect similar food aggregates.

Our study showed that three groups of DPs each contribute to the cognitive performance in AD, among which coffee/tea had the highest weighting. For protein, the lean meat and skimmed milk showed the highest weighting while the entrails and sugar also showed high impacts on MMSE. It is worth noting the high collinearity between cholesterol and high-protein food [66]. The close monitoring of the patient’s BMI and blood chemistry profiles may serve as the most appropriate measures for DP.

### 4.4. Gender Effects in Cognitive Function, DP and Lifestyle in the AD Patients

We found a gender effect in cognition, DP and lifestyle; however, the effect was affected by other collinear factors based on our statistical model. The gender differences in MMSE reflected educational-level differences between male and female, consistent with other reports [10,64,67]. Our data also suggested a male gender effect on a higher coffee/tea group DP preferences and may be related to the caregiver status. Our male patients also had higher BMI values, which may have been related to their married status and living with a spouse. If the gender effect was not directly present in AD, a proper educational plan may help to improve the care quality.

### 4.5. Factors Associated with Rapid Decline

The factors related to RCD in this study included hyperlipidemia, higher ECog score, dependent physical activity and being underweight. In contrast, the stable group was associated with independent physical activity and being obese. Age, gender, educational level, other profiles and living status were not related to RCD in our analysis. In a literature review, the presence of comorbidities [68], physical dependence [68], nutritional status [49,69,70,71], nutritional knowledge [72] and social engagement [68] were all reported to affect cognitive decline in the elderly.

### 4.6. Limitations

There are several limitations to this study. First, DP and socio-demographics showed complex interplays and the confounding factors may be hidden. It was not possible to build a statistical model covering all possible confounders based on the current data sample. In addition, the DP was assessed once in this longitudinal cohort, as we hypothesized that the DP may not change profoundly in one individual. However, it is possible that disease progression may have altered the DP in the patients or changed the caregiver behavior. Evaluating DP more than once may help to understand whether the DP changes during disease progression. Second, the DP design only investigated intake frequency of 22 food groups, while the amount of each food was not recorded. The use of frequency to assess food preference may not fully simulate the actual nutritional status or caloric intake. Currently, a few cloud-based applications (Myfitnesspal, FatSecret and Cofit) are able to estimate the amount from images, and a Taiwan Food Nutrition Ingredient Database [73] has been established to estimate the amount of nutrients. Third, there were four BMI groups in this study and a relatively small sample size of underweight group was found. The results in the underweight group may show bias and should be interpreted with caution. Finally, the RRR model for MMSE prediction showed variances. Dietary intake depends on personal food preferences. We found that only certain foods were helpful for MMSE in the AD patients; however, these foods cannot completely cover the diversity in all food choices when we provided dietary instructions. We expect to collect more food items to further confirm the relationship between food and cognitive function in AD dementia in future studies.

## 5. Conclusions

In conclusion, both modifiable and non-modifiable factors were related to cognitive decline in our AD patients. The important modifiable factors linked to DPs were BMI, lipid profile, HbA1C, B12, folate, and physical independence, and the important non-modifiable factors were age, educational levels, *ApoE4* status and caregivers.

## Figures and Tables

**Figure 1 nutrients-14-05300-f001:**
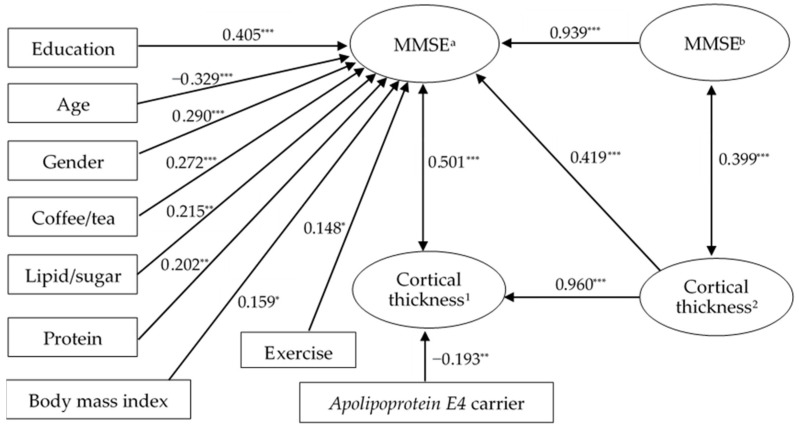
The correlation coefficients of demographic variables, three dietary patterns and MMSE. MMSE positively correlates with education, gender, 3DPs and exercise, except for age. Cortical thickness has an inverse correlation with *APOE4* but a high correlation with MMSE. However, MMSE and cortical thickness have a pair-wise high correlation. Cortical thickness represents composite scores of hippocampus, amygdala and nucleus accumbens. Cortical thickness^1^: Time point corresponds to dietary pattern data collection; Cortical thickness^2^: available historical data with the closest time point with cortical thickness^1^. MMSE^a^: time point corresponds to dietary pattern data collection, MMSE^b^: one year prior to MMSE^a^; *: *p*-value < 0.05; **: *p*-value < 0.01; ***: *p*-value < 0.001. MMSE, Mini-mental state examination.

**Figure 2 nutrients-14-05300-f002:**
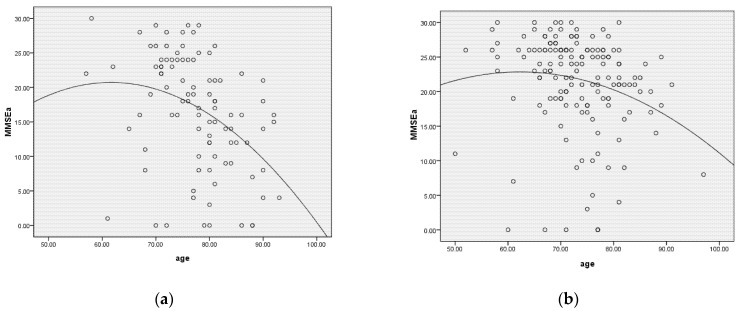
A scatter plot of age versus MMSE^a^ for the patients who lived with a spouse (*p*-value = 0.022) or not (*p*-value < 0.001). Slower decline in MMSE^a^ was found in those living with a spouse. MMSE, Mini-Mental State Examination. (**a**) The patients who did not live with a spouse; (**b**) the patients who lived with a spouse.

**Figure 3 nutrients-14-05300-f003:**
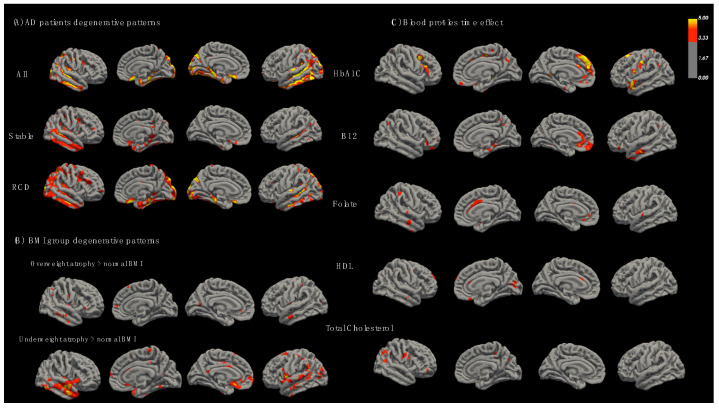
The cortical thickness changes in brain magnetic resonance imaging. (**A**) Changes in AD patients, stable and rapid cognitive decline groups. (**B**) BMI effects showing overweight or underweight BMI in contrast with normal BMI. (**C**) Significant regions related to blood chemistry profiles. BMI: body mass index. Color bar indicate log (*p*-value), parametric Gaussian-based simulations to compute the false-positive rate 0.05 and a vertex-wise threshold of 3.0.

**Table 1 nutrients-14-05300-t001:** Factor loadings for an optimal linear combination of 22 food frequencies.

Reduced Rank Regression	All (*n* = 248)		
	Factor 1	Factor 2	Factor 3
	Protein Group	Coffee/Tea Group	Lipid/Sugar Group
Lean meat	**0.717**		
Skimmed milk	**0.542**		
Beans	0.227		
Low-fat milk	0.202		
Soy products	0.177		
Oyster	0.146		
Egg	−0.057		
Fish	−0.113		
Full-fat milk	−0.137		
Octopus	−0.145		
Poultry	−0.190		
Coffee/tea		**0.990**	
Vegetable		0.132	
Fruit		−0.040	
Mushroom		−0.218	
Entrails			**0.784**
Sugar			**0.537**
Fried food			0.071
Fatty meat			−0.098
Sweet drink			−0.279
Processed food			−0.290
Dessert			−0.318
Explained variation			
Food Groups	10.084	24.789	12.744
MMSE score	4.096	7.420	4.634

Abbreviations: MMSE: Mini-mental state examination. Factor loadings for an optimal linear combination of 22 food frequencies for the three dietary patterns to explain MMSE score, derived by Reduced Rank Regression. The factor loading absolute values ≥ 0.5 are bolded.

**Table 2 nutrients-14-05300-t002:** Demographic data segregated by gender.

	All (*n* = 248)	Female (*n* = 139)	Male (*n* = 109)	*p*-Value
Age (year) ^a^	74.8 ± 7.90	74.8 ± 7.68	74.8 ± 8.22	0.711
Education (year)	7.74 ± 4.70	6.40 ± 4.59	9.45 ± 4.28	<0.001 ***
Apolipoprotein E4 carriers (*n* = 236)	68 (28.8%)	38 (29.2%)	30 (28.3%)	0.886
Body mass index, BMI (kg/m^2^)	23.99 ± 3.80	23.56 ± 4.06	24.53 ± 3.37	0.024 *
Underweight (BMI < 18.5)	17.36 ± 0.98	17.24 ± 1.00	18.05 ± 0.50	0.352
Normal (BMI 18.5~22.9)	21.03 ± 1.20	21.04 ± 1.27	21.00 ± 1.09	0.656
Overweight (BMI 23~24.9)	23.89 ± 0.61	23.71 ± 0.56	24.03 ± 0.62	0.056
Obese (BMI ≥ 25)	27.81 ± 2.69	27.88 ± 2.98	27.73 ± 2.36	0.838
Cases				
Underweight (BMI < 18.5)	14 (5.6)	12 (8.6)	2 (1.8)	0.011 *
Normal (BMI 18.5~22.9)	87 (35.1)	55 (39.6)	32 (29.4)	
Overweight (BMI 23~24.9)	54 (21.8)	23 (16.5)	31 (28.4)	
Obese (BMI ≥ 25)	93 (37.5)	49 (35.3)	44 (40.4)	
MMSE^a^ (*n* = 248)	19.51 ± 7.86	17.5 ± 8.43	22.08 ± 6.22	<0.001 ***
MMSE^b^ (*n* = 226)	20.03 ± 7.45	18.29 ± 7.94	22.25 ± 6.13	<0.001 ***
Cortical thickness^1^	0.00 ± 1.00	−0.025 ± 1.070	0.031 ± 0.912	0.432
Cortical thickness^2^	0.00 ± 1.00	0.016 ± 1.004	−0.012 ± 0.985	0.981
Everyday cognition scale (0~228)	118.6 ± 52.63	125.59 ± 53.99	109.68 ± 49.68	0.022 *
Neuropsychiatric inventory (0~144)	3.57 ± 5.52	3.73 ± 5.93	3.37 ± 4.96	0.684
Frontal behavior inventory (0~72)	8.83 ± 12.04	9.94 ± 13.09	7.42 ± 10.44	0.166
Blood data				
Glycated hemoglobin	6.13 ± 0.87	6.18 ± 0.85	6.05 ± 0.90	0.111
High density lipoprotein (mg/dL)	52.45 ± 15.02	57.18 ± 15.47	45.63 ± 11.35	<0.001 ***
Low density lipoprotein (mg/dL)	104.42 ± 34.14	107 ± 35.34	100.01 ± 32.18	0.090
Cholesterol (mg/dL)	179.37 ± 39.84	185.37 ± 42.64	170.8 ± 33.90	0.005 **
Triglyceride (mg/dL)	112.28 ± 54.16	106.65 ± 45.64	120.41 ± 63.95	0.330
B12 (pg/mL)	867 ± 637.82	951.27 ± 715.46	749.44 ± 490.41	0.029 *
Folate (ng/mL)	13.13 ± 8.28	13.92 ± 8.30	12.03 ± 8.17	0.041 *
Factor scores of 3 dietary pattern ^c^				
Protein group	0.00 ± 1.00	−0.005 ± 1.11	0.006 ± 0.85	0.414
Coffee/Tea group	0.00 ± 1.00	0.123 ± 0.99	0.157 ± 1.00	0.042 *
Lipid/Sugar group	0.00 ± 1.00	−0.034 ± 0.86	0.043 ± 1.16	0.884
Clinical Dementia Rating(CDR) ^b^				0.066
0.5	163 (65.7)	84 (60.4)	79 (73.4)	
1	56 (22.6)	32 (23)	24 (22)	
2	26 (10.5)	20 (14.4)	6 (5.5)	
≥3	3 (1.2)	3 (2.1)	0 (0)	
Rapid cognitive decline ^d^				0.757
No	171 (75.3)	95 (74.2)	76 (76.8)	
Yes	56 (24.7)	33 (25.8)	23 (23.2)	
Marital status				<0.001 ***
Married	192 (77.4)	88 (63.3)	104 (95.4)	
Widowed	51 (20.6)	48 (34.5)	3 (2.8)	
Single/divorced	5 (2)	3 (2.1)	2 (1.8)	
Self-care ability				0.055
Independent	119 (48)	59 (42.4)	60 (55)	
Dependent	129 (52)	80 (57.6)	49 (45)	
Major Caregiver				<0.001 ***
Spouse	153 (61.7)	61 (43.9)	92 (84.4)	
Others ^e^	95 (38.3)	78 (56.1)	17 (15.6)	
Comorbidity, cases (%)	247			
Hypertension	101 (40.9)	55 (39.9)	46 (42.2)	0.794
Diabetes Mellitus	55 (22.3)	32 (23.2)	23 (21.1)	0.759
Hyperlipidemia	54 (21.9)	30 (21.7)	24 (22.0)	1.00

Mann–Whitney U test for continuous variables, Fisher’s exact test for category variables. *: *p*-value < 0.05; **: *p*-value < 0.01; ***: *p*-value < 0.001. Abbreviations: MMSE, Mini-mental state examination; ^a^ Mean ± SD; ^b^ *n*; ^c^ Factor scores of 3 dietary pattern by reduced rank regression (RRR); ^d^ defined by MMSE decline ≥3 or changes of CDR score within 1-year follow-up period; ^e^ Others are defined as those who live alone, with child, relatives or in the nursing home. MMSE^a^: time point corresponds to dietary pattern data collection, MMSE^b^: one year prior to MMSE^a^; Cortical thickness^1^: Time point correspond to dietary pattern data collection, Cortical thickness^2^: available historical data with the closest time point with cortical thickness^1^.

**Table 3 nutrients-14-05300-t003:** Sociodemographic characteristics of the patients by BMI.

	Underweight ^a^	Normal ^b^	Overweight ^c^	Obese ^d^	*p*-Value	Post hoc
Sample size	14 (5.6)	87 (35.1)	54 (21.8)	93 (37.5)		
Body mass index, BMI	17.36 ± 0.977	21.028 ± 1.20	23.89 ± 0.61	27.81 ± 2.69	<0.001 ***	a < b; b < c; c < d
Blood data						
Glycated hemoglobin	5.74 ± 0.55	6.18 ± 1.14	6.02 ± 0.57	6.20 ± 0.72	0.047 *	
High density lipoprotein (mg/dL)	66.15 ± 22.93	55.29 ± 14.50	50.79 ± 14.76	48.01 ± 11.87	0.003 **	a > bcd; b > d
Low density lipoprotein (mg/dL)	127.92 ± 25.61	105.57 ± 31.44	100.36 ± 37.04	101.33 ± 35.27	0.041 *	a > bcd
Cholesterol (mg/dL)	212.77 ± 33.63	180.69 ± 36.51	174.40 ± 42.02	174.99 ± 40.62	0.008 **	a > bcd
Triglyceride (mg/dL)	93.23 ± 56.96	98.91 ± 44.34	117.44 ± 55.78	126.56 ± 58.56	0.005 **	a < d; b < d
B12 (pg/mL)	1003.77 ± 614.37	796.07 ± 449.39	861.07 ± 670.33	919.30 ± 776.53	0.560	
Folate (ng/mL)	17.54 ± 8.43	12.56 ± 8.10	13.76 ± 9.97	12.57 ± 7.20	0.018	
Everyday cognition scale (0~228)	146.21 ± 53.31	130.52 ± 58.15	118.17 ± 50.54	103.54 ± 43.87	0.003 **	a > d; b > d
Neuropsychiatric inventory (0~144)	6.07 ± 8.83	4.51 ± 6.35	3.70 ± 5.94	2.24 ± 3.03	0.11	
Frontal behavior inventory (0~72)	17.36 ± 24.76	10.8 ± 12.83	8.80 ± 10.11	5.72 ± 8.09	0.03 *	a > cd; b > d
MMSE^a^	15.50 ± 9.01	17.57 ± 8.71	21.11 ± 7.51	21.00 ± 6.42	0.005 **	a < cd; b < cd
MMSE^b^	15.93 ± 9.56	18.62 ± 7.86	21.15 ± 7.57	21.46 ± 6.03	0.022 *	a < cd; b < cd
Cortical thickness^1^	−0.697 ± 0.806	−0.175 ± 0.964	0.128 ± 1.067	0.189 ± 0.957	0.005 **	a < bc; b < d
Cortical thickness^2^	−0.924 ± 1.024	−0.062 ± 0.912	0.037 ± 1.087	0.150 ± 0.944	0.121	
Factor score of 3 dietary pattern ^e^						
Protein group	−0.626 ± 0.555	−0.080 ± 1.098	−0.029 ± 0.759	0.186 ± 1.039	0.008 **	a < cd; b < d
Coffee/Tea group	−0.185 ± 0.860	−0.091 ± 1.048	−0.072 ± 0.895	0.155 ± 1.025	0.344	
Lipid/Sugar group	−0.157 ± 0.692	−0.138 ± 0.809	0.271 ± 1.327	−0.004 ± 0.962	0.359	
Clinical Dementia Rating(CDR)						
<1	6 (42.9)	49 (56.3)	40 (74.1)	69 (74.2)	0.010 *	
≥1	8 (57.1)	38 (43.7)	14 (25.9)	24 (25.8)		
Self-care ability					0.160	
Independent	4 (28.6)	38 (43.7)	25 (46.3)	52 (55.9)		
Dependent	10 (71.4)	49 (56.3)	29 (53.7)	41 (44.1)		
Major Caregiver						
Spouse	7 (50.0)	44 (50.6)	38 (70.4)	64 (68.8)	0.028 *	
Others ^f^	7 (50.0)	43 (49.4)	16 (29.6)	29 (31.2)		

Chi-square test for category variables, Kruskal–Wallis test for continuous variables, Mann–Whitney U test for multiple comparison. *: *p*-value < 0.05; **: *p*-value < 0.01; ***: *p*-value < 0.001. MMSE, Mini-mental state examination. ^a^ Underweight, BMI < 18.5; ^b^ Normal, BMI 18.5~22.9; ^c^ Overweight, BMI 23~24.9; ^d^ Obese, BMI ≥25; ^e^ Factor scores of 3 dietary pattern by RRR. ^f^ Others are defined as those who live alone, with child, relatives or in the nursing home. MMSE^a^: time point corresponds to dietary pattern data collection, MMSE^b^: one year prior to MMSE^a^; Cortical thickness represents composite scores of hippocampus, amygdala and nucleus accumbens. Cortical thickness^1^: Time point corresponds to dietary pattern data collection, Cortical thickness^2^: available historical data with the closest time point with cortical thickness^1^.

**Table 4 nutrients-14-05300-t004:** Factors related to rapid cognitive decline in Alzheimer’s disease.

	RCD	Stable Group	*p*-Value
Sample size	56 (24.7)	171 (75.3)	
Age	74.39 ± 9.08	75.29 ± 7.55	0.922
Educational year	7.59 ± 4.86	7.69 ± 4.74	0.804
MMSE^a^	15.86 ± 8.52	20.39 ± 7.41	<0.001 **
MMSE^b^	19.00 ± 8.20	20.36 ± 7.18	0.339
Factor score of 3 dietary pattern			
Protein group	−0.03 ± 0.93	−0.03 ± 1.26	0.316
Coffee/Tea group	0.01 ± 1.00	−0.11 ± 0.96	0.342
Lipid/Sugar group	0.03 ± 1.05	−0.16 ± 0.74	0.133
Blood data			
Glycated hemoglobin	6.24 ± 1.16	6.09 ± 0.81	0.474
High density lipoprotein (mg/dL)	52.20 ± 15.64	52.46 ± 14.21	0.725
Low density lipoprotein (mg/dL)	104.98 ± 37.64	107.70 ± 34.29	0.323
Cholesterol (mg/dL)	182.71 ± 45.74	182.58 ± 39.07	0.681
Triglyceride (mg/dL)	130.05 ± 131.02	113.40 ± 55.94	0.687
B12 (pg/mL)	735.52 ± 483.32	862.76 ± 609.86	0.080
Folate (ng/mL)	11.48 ± 6.87	13.81 ± 8.50	0.100
Everyday cognition scale (0~228)	10.46 ± 13.92	8.46 ± 11.69	0.197
Neuropsychiatric inventory (0~144)	3.38 ± 6.01	3.49 ± 4.94	0.245
Frontal behavior inventory (0~72)	138.16 ± 58.47	114.72 ± 50.18	0.010 **
Gender, cases			0.757
Male	23 (41.1)	76 (44.4)	
Female	33 (58.9)	95 (55.6)	
Marital status			0.823
Married	43 (76.8)	134 (78.4)	
Widowed	12 (21.4)	34 (19.9)	
Single/divorced	1 (1.8)	3 (1.8)	
Body mass index, BMI	23.21 ± 3.68	23.95 ± 3.57	0.117
BMI, cases			0.039 *
Underweight (BMI < 18.5)	4 (7.1)	10 (5.8)	
Normal (BMI 18.5~22.9)	27 (48.2)	54 (31.6)	
Overweight (BMI 23~24.9)	8 (14.3)	44 (25.8)	
Obese (BMI ≥ 25)	17 (30.4)	63 (36.8)	
Self-care ability, cases			0.001 **
Independent	15 (26.8)	91 (53.2)	
Dependent	41 (73.2)	80 (46.5)	
Living status, cases			0.638
Spouse	33 (58.9)	107 (62.6)	
Others ^a^	23 (41.1)	64 (37.4)	
Comorbidity, cases			
Hypertension	20 (35.7)	71 (41.5)	0.530
Diabetes	14 (25)	33 (19.3)	0.447
Hyperlipidemia	21 (37.5)	30 (17.5)	0.003 **

Mann–Whitney U test for continuous variables, Fisher’s exact test for 2 × 2 cross tables. Chi-square linear association test for 3 × 2 cross tables. *: *p*-value < 0.05; **: *p*-value < 0.01. Others ^a^ are defined as living alone, with children, relatives or in the nursing home. MMSE, Mini-mental state examination. MMSE^a^: time point corresponds to dietary pattern data collection, MMSE^b^: one year prior to MMSE^a^; Cortical thickness represents composite scores of hippocampus, amygdala and nucleus accumbens. Cortical thickness^1^: Time point corresponds to dietary pattern data collection, Cortical thickness^2^: available historical data with the closest time point with cortical thickness^1^. RCD is defined by MMSE decline ≥3/year or changes of CDR score within the follow-up period.

**Table 5 nutrients-14-05300-t005:** The LASSO regression analysis of demographic variables, three dietary patterns (DPs) and MMSE score.

	Unstandardized Coefficients		z	*p*-Value	95% Confidence Interval for B	
	B	Std. Error			Lower Bound	Upper Bound
All patients (*n* = 248, AIC = 1632.2)						
(Constant)	−26.399	24.563	−1.075	0.282	−74.541	21.744
Age	1.337	0.660	2.027	0.043 *	0.044	2.630
Age*Age	−0.011	0.005	−1.96	0.049 *	−0.021	0.000
Male Gender	3.016	0.871	3.463	<0.001 ***	1.309	4.723
With Exercise habit	2.553	0.829	3.078	0.002 **	0.927	4.178
Education	0.392	0.098	4.019	<0.001 ***	0.201	0.583
Protein group	1.262	0.411	3.067	0.002 **	0.456	2.068
Coffee/tea group	0.944	0.420	2.245	0.025 *	0.120	1.768
Lipid/sugar group	0.977	0.411	2.375	0.018 *	0.171	1.783
Patients not living with a spouse (*n* = 95, AIC = 647.3)						
(Constant)	−25.183	50.748	−0.496	0.620	−124.646	74.281
Age	1.237	1.329	0.931	0.352	−1.368	3.843
Age*Age	−0.010	0.009	−1.1	0.271	−0.027	0.008
Male Gender	2.093	1.835	1.141	0.254	−1.503	5.689
With Exercise habit	2.058	1.442	1.427	0.154	−0.768	4.885
Education	0.582	0.179	3.254	0.001 **	0.232	0.933
Protein group	0.820	0.608	1.348	0.178	−0.372	2.012
Coffee/tea group	0.538	0.695	0.774	0.439	−0.824	1.900
Lipid/sugar group	0.803	0.779	1.03	0.303	−0.725	2.330
Patients living with a spouse (*n* = 153, AIC = 999.7)						
(Constant)	−22.079	29.166	−0.757	0.449	−79.244	35.086
Age	1.262	0.794	1.59	0.112	−0.294	2.817
Age*Age	−0.010	0.006	−1.588	0.112	−0.023	0.002
Male Gender	2.594	1.044	2.485	0.013 *	0.548	4.639
With Exercise habit	2.747	1.028	2.671	0.008 **	0.731	4.762
Education	0.272	0.124	2.188	0.029 *	0.028	0.516
Protein group	1.456	0.610	2.386	0.017 *	0.260	2.651
Coffee/tea group	1.049	0.538	1.949	0.051	−0.006	2.104
Lipid/sugar group	0.967	0.478	2.023	0.043 *	0.030	1.903

Response variable: MMSE, Mini-Mental State Examination. MMSE was performed at the data registration for DPs. *: *p*-value < 0.05; **: *p*-value < 0.01; ***: *p*-value < 0.001.

## Data Availability

Not applicable.

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
