# Peer review of "Cognitive Decline Related to Diet Pattern and Nutritional Adequacy in Alzheimer’s Disease Using Surface-Based Morphometry"

_nutrients, 2022, doi:10.3390/nu14245300_

Round 1

Reviewer 1 Report

Thank you for providing the opportunity to review the manuscript. The authors have linked numerous modifiable and non-modifiable factors to cognitive decline. The manuscript is well written, but there are still some concerns for the authors to consider.

Major concerns

1. I am not so sure about how modifiable and non-modifiable factors were defined. Is education non-modifiable?

2. Diet was assessed using a food frequency questionnaire. Was the food frequency questionnaire validated especially among individuals with Alzheimer’s disease? As AD patients had low memory, there might be bias in the record of dietary intakes.

3. Participants were classified as underweight, normal weight, and overweight using BMI. Since most participants were overweight, why not divide those overweight individuals into overweight and obese groups?

4. Was the distribution of the cognitive score and MRI measurements tested? If these variables were not normally distributed, the correlation analysis may not be suitable.

5. Complex findings have been provided by the research, but it is hard for the readers to catch the main information the authors are aiming to deliver.

Minor concerns

1.     Table 3: The small number of underweight participants might result in bias.

2.     Figure 1: Was BMI analyzed as continuous variable?

3.     Figure 2: The tiles of the axis were not well organized.

4. How were missing values treated?

Author Response

Dear Reviewer 1, we have revised according to your point and the revised contents were labelled with blue color for your reference. Thanks for your comments.

Major concerns

Point 1: I am not so sure about how modifiable and non-modifiable factors were defined. Is education non-modifiable?

Response 1: We agree with the reviewer that education may be considered as a modifiable factor in certain circumstances. The rationale that we considered education as a non-modifiable factor in our study was based on the disease our patients had. As we enrolled patients with AD in this study, we considered their educational levels may not change at the time we surveyed. Therefore, we reported the educational year as non-modifiable factor in this study. We have included the rationale in this revision on page 3, line 127~128.

Point 2: Diet was assessed using a food frequency questionnaire. Was the food frequency questionnaire validated especially among individuals with Alzheimer’s disease? As AD patients had low memory, there might be bias in the record of dietary intakes.

Response 2: In this study, the food frequency questionnaire was collected from the major caregiver’s observation rather than the patients. The response had been included on page 3, line 135~136.

Point 3: Participants were classified as underweight, normal weight, and overweight using BMI. Since most participants were overweight, why not divide those overweight individuals into overweight and obese groups?

Response 3: According to the reviewer’s suggestion, we have divided the overweighted group into overweighted (23-24.9 kg/m2), and obese (BMI ≥25 kg/m2) groups. The comparisons between overweighted and obese groups were not significant in blood data, cognitive performances, cortical thickness or factor loading. The revised group stratification was included on page 3, line 130~131, Table 2~4 (page 6-9) and supplementary Table 3.

Point 4: Was the distribution of the cognitive score and MRI measurements tested? If these variables were not normally distributed, the correlation analysis may not be suitable.

Response 4: The cognitive score and MRI measurements were not normally distributed. Based on the Central Limit Theorem in Statistics, we used correlation analysis to explore the relationships among continuous variables. As our sample size was large, the analysis may detect the significance.

Point 5: Complex findings have been provided by the research, but it is hard for the readers to catch the main information the authors are aiming to deliver.

Response 5: The relationships among food frequency questionnaire, cognition, surface morphometry and demographic data were complex, so we reported the results consequently in different result sections. A summary of our results can be found in Figure 1 (on page 10, line 266) and in the discussion section 1 (4.1 major finding, page 12, line 309). From Figure 1, the significant factors were selected based on the sequential statistical analysis and entered in the correlation analysis to show their relationships with MMSE and surface morphometric data. From the major finding (4.1 major finding), we reported significant modifiable and non-modifiable factors related to cognitive test scores in AD. The composite surface morphometry of hippocampus, amygdala, and nucleus accumbens can be considered as endophenotypic regions for estimating cognitive performances in AD. Lastly, we reported underweighted group may be associated with faster cortical thickness atrophy, accounting for the group effects of rapid cognitive decline.

Minor concerns

Point 1: Table 3: The small number of underweight participants might result in bias.

Response 1: Indeed, the small sample size might result in bias. In this revision, we still reported the finding of the underweight group, but we also added the limitation to remind the reviewer (page 14, line 421~423). Although small samples may result in bias, we used the nonparametric analysis methods in Table 3 and Table S3 that are considered suitable for small samples.

Point 2: Figure 1: Was BMI analyzed as continuous variable?

Response 2: Yes, the BMI was analyzed as a continuous variable.

Point 3: Figure 2: The tiles of the axis were not well organized.

Response 3: Thank you for the comment. The titles of the y-axis were corrected as MMSEa in Figure 2 (page 11, between line 285 and 286).

Point 4: How were missing values treated?

Response 4: We excluded observations with missing values when the variables were used in each analysis, so we used the most complete data. Thank you for the comment.

Reviewer 2 Report

A relevant study, that explored the reflect of demographic and metabolic factors in the Alzheimer’s disease progression.  

This study has some important points, emphasizing the patient’s number (248) and the utilization of neurobehavioral in addition to cortical measurements derived from magnetic resonance images.

Some results, related with dietary pattern (BMI and coffee/tea), complement the scientific literature showing important cognitive protective factors.   

Author Response

Response to Reviewer 2 Comments

Point 1: A relevant study, that explored the reflect of demographic and metabolic factors in the Alzheimer’s disease progression.

This study has some important points, emphasizing the patient’s number (248) and the utilization of neurobehavioral in addition to cortical measurements derived from magnetic resonance images.

Some results, related with dietary pattern (BMI and coffee/tea), complement the scientific literature showing important cognitive protective factors.

Response 1: Thanks for the response from Review 2.